# Evaluation of Health Promotion in International Schools Using the Schools for Health in Europe (SHE) Rapid Assessment Tool

**DOI:** 10.3390/healthcare13060633

**Published:** 2025-03-14

**Authors:** Jaime Barrio-Cortes, María Díaz-Quesada, María Martínez-Cuevas, Amelia McGill, Cristina María Lozano-Hernández, Cayetana Ruiz-Zaldibar, María Teresa Beca-Martínez, Montserrat Ruiz-López

**Affiliations:** 1Faculty HM of Health Sciences, University Camilo José Cela (UCJC), 28660 Madrid, Spain; amcgill@ucjc.edu (A.M.); cristina.lozano@uclm.es (C.M.L.-H.); crzaldibar@ucjc.edu (C.R.-Z.); mariateresa.beca@ucjc.edu (M.T.B.-M.); mrlopez@ucjc.edu (M.R.-L.); 2Health Research Institute of HM Hospitals, 28015 Madrid, Spain; 3Foundation for Biosanitary Research and Innovation in Primary Care (FIIBAP), 28003 Madrid, Spain; 4Research Network on Chronicity, Primary Care and Prevention and Health Promotion, Carlos III Health Institute, 28029 Madrid, Spain; 5Santa Isabel SEK International School, SEK Institution, 28012 Madrid, Spain; maria.diaz@sek.es; 6Fuencarral Healthcare Centre, Madrid Health Service, 28034 Madrid, Spain; mmartinezcuevas@salud.madrid.org

**Keywords:** health promotion, health-promoting schools, participatory health promotion, public health strategies, rapid assessment tool

## Abstract

**Background:** Many schools are committed to the “Five Steps to a Health Promoting School guide” created by the Schools for Health in Europe (SHE) network to avoid chronic disease and promote healthy environments. **Objectives:** The aim of this study was to evaluate schools’ health promotion policies and practices via the SHE rapid assessment tool. **Methods:** From February 2019 to June 2019, a cross-sectional survey based on this tool was conducted in nine international schools that are part of a private network operating in multiple countries. The entire school community was offered the opportunity to participate. The survey consisted of 37 questions aligned with the whole-school approach to health promotion and addressed orientation, healthy school policy, the physical and social environment, health skills, community links, and staff well-being using a three-point scale. A univariate analysis was subsequently performed. **Results:** A total of 929 people responded, 59.7% of whom were women, representing 74 different nationalities (82.5% Europeans). The average age was 25.9 years, ranging from 7 to 77 years. The participants included 57.2% students, 21.3% teachers, 15.3% families, 3.2% management teams, 1% counsellors/psychologists, 1% administrative/service staff, 0.5% catering staff, 0.3% nursing/medical staff, and 0.2% maintenance staff. The findings revealed that 80% of the respondents felt that most of these health promotion needs were being partially or fully addressed, predominantly with medium or high priority. However, 30% of the respondents indicated that the school had not yet assessed the students’ and employees’ health and well-being needs or fully promoted a healthy work-life. **Conclusions:** The SHE rapid assessment tool serves as an effective initial step in identifying key indicators within the school community, offering insights for future action towards becoming a health-promoting school. These results underscore the importance of addressing social and behavioural determinants of health within different international educational settings to promote positive sociorelational well-being and youth development. By fostering the well-being of children, adolescents, and the educational community, schools play a pivotal role in reducing the risk factors for chronic diseases and supporting psychosocial adaptation.

## 1. Introduction

Children from diverse socioeconomic, ethnic, and cultural groups spend significant hours in schools [1,2]. Education attendance is mandatory and accessible for individuals aged 4 to 16 years in many regions across the globe [3]. This phase of life is characterised by high receptivity, facilitating rapid acquisition of knowledge and habit formation [4,5]. Despite educational efforts, external influences, such as family dynamics, media exposure, peer pressure, and socioeconomic and cultural factors, can adversely affect health, with evidence suggesting that poor health can hinder learning [6]. Family and community involvement play crucial roles in mitigating these negative influences, contributing significantly to knowledge exchange. Therefore, schools are ideal settings for fostering healthy and sustainable environments while encouraging lifestyle habits that promote better overall health and help prevent chronic diseases [7,8,9,10,11].

Health-promoting schools (HPSs) have been recognised as strategic vehicles for promoting positive educational development and healthy behaviours, aligning with the United Nations’ goals of health for all and education for all [12]. The European Network of Health Promoting Schools (ENHPS), established in the 1980s, stands as a thriving practical example of the health promotion movement. It began with the participation of seven countries (Austria, Belgium, Finland, Germany, the Netherlands, Portugal, and Sweden) [13]. However, since 1991, it has been integrated into the school curricula through the collaboration of the European Commission, the Council of Europe, and the World Health Organization Regional Office for Europe (WHO Europe), with its membership now spanning 43 countries [14].

The ENHPS was renamed in 2009 and became known as the Schools for Health in Europe (SHE) network. This transition aimed to empower the school community to exert greater control over its health and environment within the broader European region [15]. The SHE network established a valuable framework for schools, aiming to increase their capacity for sustainable and holistic health prevention and promotion efforts, ultimately transitioning them into HPSs [16]. Local, regional, national, and international objectives are considered in the development of health promotion indicators [9,10,17,18].

The SHE network health promotion approach was built upon five pillars: participation, school quality, evidence, school community, and the whole-school approach to health [19]. It operates on five values: equity, sustainability, inclusion, empowerment, and democracy [19]. These strategies seek to transform the current conditions to improve the overall health and quality of life and reduce the risk of chronic diseases. The ambition of this project fosters community participation by involving students, families, teaching and nonteaching staff, health services, meeting spaces, and media in activities and decision-making processes outside the traditional education curriculum [20,21].

The “SHE Online School Manual 2.0” offers a methodological guidebook for becoming an HPS, assisting school administrators, teachers, and health and education personnel. Structured into five distinct phases, this manual serves as a practical roadmap to navigate the process effectively. In the first phase, schools define their goals and objectives for the health-promoting school initiative. The second phase involves evaluating the school’s current status via the SHE rapid assessment tool, which identifies strengths, areas for improvement, and necessary actions. This evaluation establishes a baseline and highlights existing health policies, including those targeting chronic disease risk reduction. The tool also supports ongoing assessment and progress adjustments. The tool is valuable for schools because it considers policies, practices, and environmental factors; helps identify areas for improvement; and tracks progress. The tool is recommended internationally as it aligns the schools with global health promotion standards and fosters a healthier school environment. The third phase focuses on developing action plans based on assessment findings, outlining strategies and interventions to address priorities, such as preventing chronic diseases through healthier environments and practices. The fourth phase implements these plans, which require stakeholder engagement and resource allocation for health promotion activities. Finally, the fifth phase involves evaluating and monitoring implemented actions by gathering data, analysing results, and assessing their impact on the health and well-being of students and staff. Monitoring systems track success, particularly in addressing lifestyle-related chronic disease risks, and allow for necessary adjustments to enhance effectiveness [20,22,23].

This study fills a gap in the literature, as while the SHE rapid assessment tool is widely used to evaluate school health promotion policies, no previous studies have analysed its results within a school community. The tool has mainly been used for needs assessment, whereas this study explores its practical application in real-world settings.

The main objective of this study was to assess the current health promotion policies and practices within a network of international schools via the SHE rapid assessment tool. The rationale for conducting this research within this school network is the collaboration between these educational centres and a university’s school health program. This study engaged the entire educational community in identifying and addressing health needs and priorities, with a focus on promoting health and preventing chronic diseases. This research addresses the current health promotion policies and practices in a network of international schools, examining how they align with global health promotion standards. It explores the strengths and areas for improvement in these initiatives and identifies ways to enhance these efforts to better meet the needs of the school community.

## 2. Materials and Methods

### 2.1. Study Design and Sampling

A descriptive cross-sectional study with an analytical approach was developed, involving an estimated population of over 12,000 individuals within the SEK International Schools educational community. The survey targeted diverse groups, including students, families, and both teaching and nonteaching staff from schools located in multiple countries. Since the study took a whole-school approach, the responses reflect the perspectives of the entire school community rather than distinguishing between specific groups. The participating schools included SEK International School Ciudalcampo, SEK International School El Castillo, and SEK International School Santa Isabel in Madrid, Spain; SEK International Catalonia in Barcelona, Spain; SEK International Atlántico in La Coruña, Spain; SEK International Alborán in Andalucía, Spain; SEK International School Qatar in Doha, Qatar; SEK International Les Alpes in Flumet, France; and SEK International School Dublin in Dublin, Ireland [24].

### 2.2. Data Collection

Data were collected through an online questionnaire. The survey targeted students between 6 and 18 years of age, teaching staff, nonteaching staff, and families within the SEK educational community who were able to understand the questions. The data collection period spanned from February 2019 to June 2019. All potential participants received the survey link via internal institutional email. Additionally, parents could access the questionnaire through a link provided on the school’s online platform designed to keep families informed about their children’s school life, ensuring that all members of the community had the opportunity to participate with support from school nurses and other staff. Participation in the survey was voluntary and anonymous.

### 2.3. Variables

The independent variables collected were sex, age, nationality, and relationship with the school. The dependent variables were questions collected from the SHE rapid assessment tool recommended by SHE network, which comprises a series of 37 questions aligned with the whole-school approach to health promotion; these are divided into distinct spheres corresponding to the evaluation elements [20]. These elements include orientation, healthy school policies, the physical environment, the social environment, health skills, links to the community, and healthy school teams. All the questions are listed in Appendix A. Each question was answered based on priority and the current situation using a three-point scale:Current—the school’s current situation on a three-point scale:1 = not in place.2 = partly in place.3 = fully in place.4 = don’t know.Priority—the priority that the school community gives to the question on a three-point scale:1 = low/no priority.2 = medium priority.3 = high priority.4 = don’t know.

This approach ensures clarity and simplicity, allowing schools to quickly assess and identify areas for improvement without unnecessary complexity. The results of the assessment can be interpreted by considering the score for each question with respect to the current situation and priority level. Areas with a low score on the school’s current situation and a high score on priority can be the focus of future action in the process of becoming an HPS. The three-point scale aligns with the tool’s purpose of providing a rapid and effective evaluation of health promotion initiatives, making it a practical choice for schools in their assessment process.

At the end of the survey, the participants were given an opportunity to share their thoughts freely through an open-ended question. This allowed them to provide any comments, suggestions, or reflections that they deemed relevant. The open-ended responses were analysed via a whole approach, where the responses were categorised based on common themes or topics related to health promotion. The categorisation process was guided by predefined criteria aligned with the study’s objectives, and the responses were classified into relevant categories to facilitate quantitative analysis. This ensured consistency and clarity in interpreting the open-ended data while maintaining the focus on key themes related to the research question.

### 2.4. Data Analysis

An initial phase of database debugging was conducted, followed by a subsequent tabulation analysis. Statistical univariate analysis of independent and dependent variables was performed using the mean and standard deviation (SD) for quantitative variables that followed a normal distribution. For variables that did not follow a normal distribution, the median, minimum, and maximum values were used. Measures of frequency and percentages were employed for qualitative variables. A 95% confidence interval was calculated for the analysed data. The analysis was performed via the IBM SPSS statistics v.25 statistical program.

## 3. Results

A total of 929 (7%) participants responded to the survey. Among them, 59.7% were women, and their ages ranged from 7 to 77 years, with an average age of 25.9 years (standard deviation = 16.3). The respondents represented 74 different nationalities globally, with the majority (82.5%) being European. Each respondent had a distinct relationship with the SEK international schools, with 57.2% being students, 21.3% being teachers, 15.3% being parents, 3.2% being part of the management team, 1% being social workers and psychologists, 1% being administration and services staff, 0.3% being catering staff, 0.3% being medical staff, and 0.2% being maintenance personnel (see Table 1).

With respect to the questions related to orientation in the SHE rapid assessment tool, almost all points were reported as either totally or partially currently in place, except for Question 1.4. In this question, respondents were asked whether the school has evaluated the needs and desires of the students and staff regarding health and well-being. Over one-third indicated that this aspect was not currently achieved. In terms of priority, almost all points were considered high priority in 50% or more of the cases, with 25–30% of the points being considered medium priority (Figure 1).

Concerning questions related to the healthy school policy, most were reported as either totally or partially currently in place in more than 75% of the cases. However, in almost 25% of the cases, questions related to students, teaching/nonteaching staff, and parents’ participation in the school’s planning and implementation of health-related activities were not currently in place. In terms of priority, all points were considered high priority in more than 40% of the cases (see Figure 1).

With respect to the questions related to the schools’ physical environment, many points were reported as either totally or partially currently in place in approximately 75% of the cases. However, in more than 25% of the cases, the questions related to students and staff having access to school facilities for physical activity outside of school hours and the safety and design of the route to the school to encourage students to engage in physical activity were not in place. In terms of priority, almost all points were considered medium or high priority in more than 80% of the cases (Figure 2).

Concerning the questions related to the schools’ social environment, in nearly 50% of the cases, the majority of points were reported to be fully in place. Similarly, in terms of priority, all points were considered high priority in more than 50% of the cases (Figure 2).

In terms of health skills, almost 80% of the items were considered fully or partly in place, whereas more than 50% were deemed high priority (Figure 3).

With respect to the questions related to community links, all the items were reported as either partly or totally in place in more than 75% of the cases. In terms of priority, all points were considered high priority in more than 40% of the cases, with more than 60% indicating that a school with clear rules that promote healthy behaviours was a high priority (Figure 3).

In terms of priority, all points were given high priority in more than 40% of the cases (Figure 3).

Finally, in the questions related to healthy school staff, the survey revealed that most objectives were reported as either completely or partially achieved. However, in Question 7.3, “Our school promotes a healthy work-life balance and a reasonable workload and provides an open environment to discuss work-related problems and stress”, 35% of the respondents believed it was not achieved. In terms of priority, most of the respondents considered it a very high priority (Figure 3).

In Figure 4, a spider chart displays the averages of the different responses of the questions within each sphere of the SHE rapid assessment tool. Regarding the ‘current (in place)’ status, responses generally fall between ‘partly’ and ‘fully’ implemented. On the other hand, for ‘priority’, most responses indicate ‘high’ or ‘medium’ importance across all seven spheres.

A detailed breakdown of the participant demographics—separately for students, teachers, parents, the management team, social workers or psychologists, administration and services staff, catering staff, nurses and doctors, and maintenance staff—is provided in Appendix A.

A total of 245 participants shared their opinions at the end of the survey. Most of these comments either suggested improvements or praised the school. Among them, the most prevalent topics were related to the environment, school facilities, and teachers, with additional comments focusing on physical exercise and food (Table 2).

## 4. Discussion

### 4.1. Main Findings

The survey gathered responses from 929 participants across nine international schools, providing valuable insights into health promotion within an international school network. The diverse sample included students, teachers, parents, and staff, representing a wide range of nationalities, with the majority being European. This broad group provided a collective perspective of the entire school community without distinguishing between specific stakeholder groups.

The study revealed that this network of international schools has made significant progress in implementing health promotion initiatives. The responses to the SHE rapid assessment tool indicated that most health promotion points were either fully or partially in place, demonstrating substantial efforts to integrate health practices into the school environment. Schools showed a strong commitment to supporting the health and well-being of students and staff, with a particular focus on mitigating the risk factors associated with chronic diseases. However, areas such as evaluating the health needs and desires of the school community still require further attention.

Most respondents considered health promotion initiatives to be of high or medium priority, reflecting the importance of addressing health-related issues within the school network. This collective prioritization emphasizes the critical role that schools play in promoting physical activity, healthy eating, and mental well-being. These efforts help prevent and manage chronic diseases, showing the proactive steps schools took to support healthier lifestyles and create supportive environments for their communities.

Despite these positive outcomes, areas in need of improvement were identified. Concerns about staff well-being emerged, particularly regarding a healthy work-life balance. Additionally, gaps were noted regarding participation in health-related activities and access to physical activity outside of school hours. Open-ended feedback from the participants echoed these concerns, with common themes related to enhancing the physical environment, improving school facilities, and promoting better food and exercise options for students and staff.

### 4.2. Characteristics of the Population

The survey attracted a varied range of participants from across the global educational community, including individuals of various age groups, sexes, and nationalities.

Nearly 60% of the participants were women. In contrast, another study on the SHE network reported a slightly higher proportion of male participants (52%) in Spain [25], whereas in Finland, female participants were more prevalent (56.5%) [26]. This study’s broad age range—including students, teachers, and families—demonstrates a diverse respondent group across different generations. This contrasts with previous studies regarding HPSs, where only the students aged 8 to 18 were included. This inclusive approach provides a more holistic view of health promotion within the school community, offering insights that may differ from those based solely on a student-focused sample [25,26,27].

The significant presence of 74 nationalities within international schools located in Spain, France, Qatar, and Ireland, with a prevalent European presence, highlights the study’s global reach and the potential for diverse cultural perspectives. Furthermore, the breakdown of the respondents’ roles within this network of international schools highlights the multistakeholder nature of the study, with significant input from students, educators, parents, and both teaching and nonteaching staff.

### 4.3. Health Promotion Assessment

The HPS concept has been shown to improve students’ health and well-being while also supporting their education and learning within the educational settings. Successfully implementing an HPS framework entails a multifaceted and innovative approach across various domains, such as the curriculum, school environment, and community involvement. The whole-school approach of the HPS concept represents a crucial initiative in fostering health within school settings. Numerous studies have validated its effectiveness across a range of school-based programs, particularly in promoting healthy behaviours, such as balanced nutrition, regular physical activity, and mental well-being—key factors in preventing chronic diseases [28,29,30]. To implement the HPS framework effectively, schools need to assess and evaluate the results in areas like orientation, policies, environments and community engagement. These indicators could serve as strategies to implement changes to promote health and education. The data from the SHE rapid assessment tool act as a starting point that will allow future planning for schools to become an HPS [13,17,20].

Our findings from the SHE rapid assessment tool revealed a generally positive implementation of orientation-related aspects within the schools. Furthermore, the analysis delves into the priority ranking of these orientation-related elements. While many of the respondents regarded almost all points as high priority, a subset ranked them as medium priority. These differences in perceptions highlight the need for targeted efforts to better understand and address health priorities within the school community. Including stakeholders’ perspectives is essential in developing and evaluating the health promotion programs. As recommended in other studies, schools can enhance engagement by conducting regular surveys to gather feedback from all the stakeholders, organizing focus groups to ensure a deeper understanding of varying perspectives, and creating opportunities for open discussions about the importance of health promotion activities. Additionally, tailored professional development for staff can help align health priorities, while actively involving students and parents in the decision-making processes can ensure a more comprehensive and inclusive approach to health promotion [31,32,33,34].

Recognizing the variability in priority perceptions underscores the importance of involving stakeholders in discussions to align priorities with the collective needs and aspirations of the school community [35]. Such insights could inform strategic interventions aimed at enhancing the overall health and well-being landscape within the school community. A notable exception arises in the question of whether a school has assessed the health and well-being needs and desires of its students and staff. Here, more than one-third of the respondents indicated that this aspect was not currently achieved, suggesting potential gaps in understanding the health priorities of the school community. This finding is consistent with other studies showing that the opinions of students and staff with respect to their needs are often overlooked when planning health promotion initiatives [36].

Similarly, the findings related to the Healthy School Policy reveal areas for improvement, particularly concerning the involvement of students, teaching/nonteaching staff, and parents in health-related activities. While empirical literature on the involvement of stakeholders in school policy development and implementation is limited, it is evident that engaging the stakeholders plays a crucial role across the entire policy continuum [37]. Enhancing the participation and engagement of these stakeholders could contribute to the effectiveness and sustainability of health promotion efforts within schools, as has also been demonstrated in other studies [11,38,39,40,41].

In terms of the physical environment, while most points were reported as currently in place, certain areas required attention, such as access to school facilities for physical activity outside of school hours and safety measures to encourage physical activity. Addressing these issues, as proposed in the existing research, can help foster environments that better support healthy behaviours among students and staff [9,11,42].

The positive feedback regarding the social environment, with school facilities promoting enjoyment and sociability, underscores the importance of creating supportive and inclusive environments for students. This aligns with the literature, which states that a school’s social environment refers to the quality of relationships among the students, teachers, staff, and leaders, with key elements including well-being, safety, inclusion, diversity, and the impact of social media and virtual settings [20,43]. Additionally, evidence suggest that structured interventions, such as music therapy, can significantly enhance the social interaction and communication skills, particularly in children with developmental challenges, such as autism spectrum disorder [44]. Implementing such approaches within school settings may further support social inclusion and student well-being.

Prioritizing the establishment of healthy physical and social school environments is crucial, as many studies have illustrated their positive impact on the physical activity, dietary habits, and mental well-being of both students and employees [45]. However, there is a need for more detailed studies to strengthen the evidence supporting the feasibility and effectiveness of interventions aimed at community/relationship building, empowering student involvement in modifying school food and physical activity environments, and upgrading school facilities [46].

With respect to the health skills, most items were fully or partially implemented, which demonstrates a substantial level of progress in this area. This high percentage suggests that this network of international schools was actively working towards integrating these health skills into their curriculum and practices. However, more than half of these items were also considered high priority, indicating a recognition of their importance and perhaps an urgency to address them more comprehensively. Interestingly, while previous studies have reported positive associations between HPSs and life skills, lifestyle habits, and academic performance, recent findings suggest that the impact of these health programs on students’ health and academic performance may be limited [47]. Therefore, schools are encouraged to conduct a thorough evaluation of their health programs and consider integrating more comprehensive and evidence-based interventions.

The findings on community links emphasize the importance of building strong connections between schools and their surrounding communities. A key aspect of achieving this lies in fostering partnerships and collaborations among different sectors at the national, regional, and local levels, while engaging everyone involved in the daily life of schools. These stakeholders include school management, teachers, nonteaching staff, parents, and most importantly, the students themselves. Involving this diverse community in the design and implementation of programs helps ensure they are more effective and better suited to the specific needs of the school community [27,48].

Finally, while most goals for supporting healthy school staff were met, over a third felt that promoting work-life balance and managing stress was not adequately achieved, despite being prioritized. This highlights a key gap in staff well-being initiatives. Other studies support this need for schools to prioritize comprehensive strategies that address staff health, reduce workplace stress, and foster more supportive and positive working environments [49,50].

Overall, the findings underscore the importance of prioritizing health promotion in educational settings and highlight the positive impact that proactive health promotion initiatives can have on the well-being of both students and staff [51]. The successful implementation of HPS programs has demonstrated that their principles and methods could significantly improve the health and well-being of students and communities. These initiatives also enhance the educational experience for all young people within these environments [52], while contributing to the prevention of chronic diseases by addressing key risk factors, such as unhealthy diets, sedentary behaviours, and stress.

This study is designed to reflect a collective perspective on health promotion within the school community, following a whole-school approach [20]. This approach integrates strategies that promote healthier lifestyles and support the early identification and mitigation of chronic disease risk factors, creating a more sustainable foundation for long-term health and well-being. Despite this whole-school approach, differences in perspectives among the stakeholder groups emerge, providing deeper insights into how health promotion is perceived and implemented across the school community.

### 4.4. Strengths and Limitations

The descriptive nature of this study limits its ability to establish causality and temporality and precludes a comparative analysis. Future comparative or longitudinal studies could further strengthen the insights in this research area. Data were collected through voluntary surveys, which, while offering valuable insights, may not provide the same depth as interviews. This introduces the possibility of response bias, where the design of the survey or the way questions are framed may unintentionally encourage certain types of responses, leading to measurement errors.

Moreover, selection bias may have influenced the results, as participation was limited to a specific segment of the group, such as highly engaged individuals, which could result in an overestimation of the perceived success of health promotion initiatives.

To address the nonprobabilistic sample, the study included participants from diverse demographics, capturing a broad range of perspectives on health promotion. This study offers valuable insights into health promotion in an international school network based on a diverse sample. However, given the low response rate (7%), the results may not fully represent the entire school community.

Additionally, a notable limitation of the study is the lack of publicly available reliability data for the SHE rapid assessment tool, which may impact the strength of the conclusions drawn from the results.

While this study was conducted in 2019, the challenges observed have persisted to the present day, as the situation following the COVID-19 pandemic has not significantly changed in terms of health promotion in educational settings.

Overall, despite these limitations, the methodology provided an in-depth exploration of the health promotion challenges in educational settings. Furthermore, this study is the first to use the SHE rapid assessment tool in a research context, even though it is widely used for evaluating school health promotion. This innovative approach provides valuable data that can inform future health promotion strategies.

## 5. Conclusions

This study highlights the value of the SHE rapid assessment tool in evaluating health promotion school initiatives in schools, identifying both strengths and areas for improvement. The survey, conducted across nine international schools with 929 participants from diverse backgrounds, provided a comprehensive perspective on health promotion within an international school network.

Findings indicate significant progress in integrating health promotion practices, with most assessed areas either fully or partially implemented. Schools demonstrated a strong commitment to student and staff well-being, particularly in mitigating chronic disease risk factors. However, a systematic evaluation of the community health is needed and improvements in staff well-being, work-life balance, and access to extracurricular activities remain areas for further development.

Addressing challenges such as physical inactivity, unhealthy eating, and stress can help schools foster resilience, reduce psychosocial stress, and prevent issues like stigmatisation, marginalisation, and bullying. By engaging diverse perspectives, schools can further strengthen their health strategies, address specific community needs, and improve overall participation.

Enhancing stakeholder participation, integrating chronic disease prevention strategies, and using the SHE tool to address these gaps will contribute to creating healthier and more supportive school environments.

By taking a proactive approach, schools not only strengthen physical health but also cultivate an inclusive, supportive atmosphere that promotes long-term well-being and positive social development.

## Figures and Tables

**Figure 1 healthcare-13-00633-f001:**
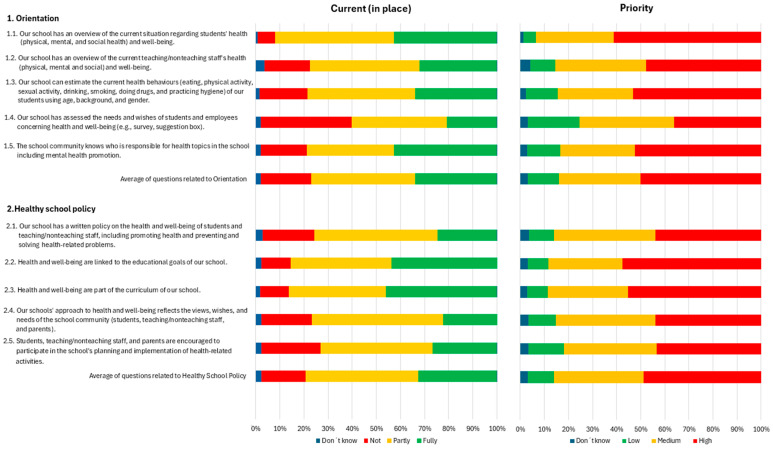
Questions related to orientation and healthy school policy.

**Figure 2 healthcare-13-00633-f002:**
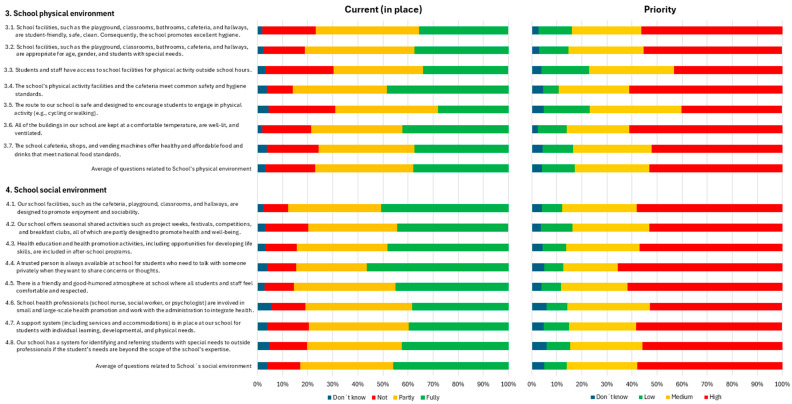
Questions related to the school’s physical and social environment.

**Figure 3 healthcare-13-00633-f003:**
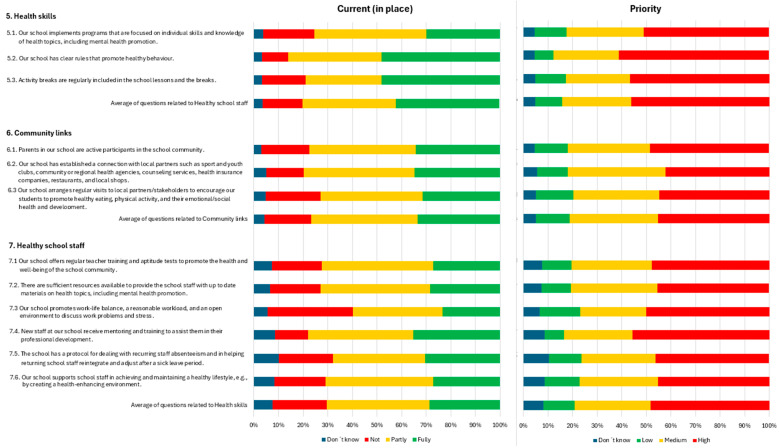
Questions related to health skills, community links, and healthy school staff.

**Figure 4 healthcare-13-00633-f004:**
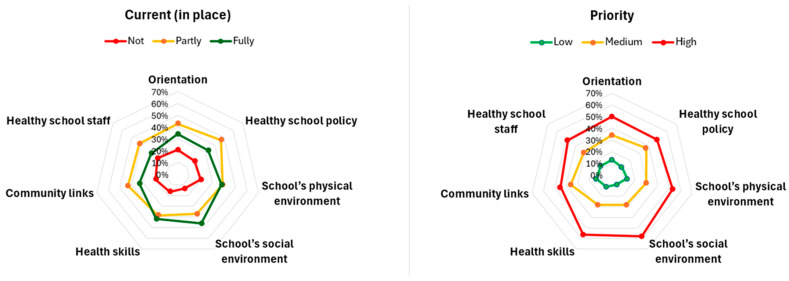
Averages of the different responses to the questions within each sphere of the Schools for Health in Europe Rapid Assessment Tool.

**Table 1 healthcare-13-00633-t001:** Characteristics of the respondents.

Sociodemographic Characteristics n (%)	Total Sample 929 (100)	95% Confidence Interval
Sex		
Male	374 (40.3)	37.1–43.4
Female	555 (59.7)	56.6–62.9
Age (years)		
Mean (Standard deviation)	25.9 (16.3)	24.9–27.0
Median (Interquartile range)	18 (12–43)	
Relationship with the school		
Students	531 (57.2)	54.0–60.3
Teachers	198 (21.3)	18.7–23.9
Parents	142 (15.3)	13.0–17.6
Management team	30 (3.2)	2.1–4.4
Social worker or psychologist	9 (1)	0.4–1.6
Administration and services staff	9 (1)	0.4–1.6
Catering staff	5 (0.5)	0.1–1
Nurses and doctors	3 (0.3)	0.1–0.6
Maintenance staff	2 (0.2)	0.01–0.5
Continents nationality		
Europe	766 (82.5)	80.0–84.9
America	75 (8.1)	6.3–9.8
Asia	68 (7.3)	5.6–8.9
Africa	17 (1.8)	1–2.7
Oceania	3 (0.3)	0.1–0.5

**Table 2 healthcare-13-00633-t002:** Feedback from open-text responses.

Open-Text Responses n (%)	245 (100)
Related to the environment/school/teachers	66 (27.0)
Related to physical exercise and food	53 (21.6)
Related to SQFR (Suggestions, Questions, Feedback, and Requests) regarding the questions in the quick assessment tool.	28 (11.4)
Related to emotional well-being	26 (10.6)
Related to healthy leisure	26 (10.6)
Related to affective sexual education	2 (0.8)
Related to other reasons	44 (18.0)

## Data Availability

Datasets generated and analysed during the current study are available from the corresponding author upon reasonable request.

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
