# Peer review of "Evaluation of Health Promotion in International Schools Using the Schools for Health in Europe (SHE) Rapid Assessment Tool"

_healthcare, 2025, doi:10.3390/healthcare13060633_

Round 1
Reviewer 1 Report
Comments and Suggestions for Authors
Dear Authors,
Congratulations!
The article “Health Promotion Evaluation in International Schools Using the Schools for Health in Europe Rapid Assessment Tool” addresses a relevant and current theme, and presents an appropriate methodological design and data analysis. Overall, the manuscript presents a good level of scientific quality, but there are some points that require improvement.
In the introduction, more recent references should be added to the literature review to strengthen the argument. The argument regarding the relevance of the topic under study should also be developed.
It would be important to add information about the validation and limitations of the tool used for data collection.
“In this study, nearly 60% of participants were women.” After this sentence it will not be necessary to add: “In contrast, men, comprising approximately 40% of the participants.” (line 250-251)
“This contrasts with other studies regarding with HPS, where only students aged between 8 and 18 years old were asked [20–22].” (line 255-256) It would be relevant to discuss the implications of this sample characteristic on the study results.
(see also the impacts of some biases, of the profile of the participants, on the results obtained: about the fact that more women responded, about the response rate being 7%...)
“The notion of a HPS has proven effective in enhancing the health and well-being of students, while also aiding teaching and learning within educational settings.” (line 264-265) I don't understand the meaning of the sentence! It should be rewritten for better understanding…
“The data from the SHE rapid assessment tool act as a starting point which will allow future planning for organizing to become a HPS [12, 16, 18].” (line 275-277) The sentence seems out of place to me!! It's not a discussion!
A review of the discussion is recommended in order to avoid repetition of ideas and improve the fluidity of the text. The comparison of results with other studies should be improved and deepened to reinforce the reading and interpretation of the results obtained.
Finally, it seems important to highlight more clearly the practical implications of the study carried out and suggest recommendations for future research.
I hope that my notes can contribute to the improvement of the presented manuscript.
Comments on the Quality of English LanguageIt would be important to review the English used in this article to allow for a more fluid reading of the text.
Author Response
Dear Reviewer 1,
Point-by-Point Response to Reviewer 1's comments:
Quality of English Language
The English could be improved to more clearly express the research.
Response: We appreciate your feedback. To ensure the highest quality and clarity in expressing the research, we have sent the text to American Journal Experts (AJE) for professional editing. We believe this has helped to refine the language and ensure the research is communicated accurately and effectively.
Comments and Suggestions for Authors
Dear Authors,
Congratulations! The article “Health Promotion Evaluation in International Schools Using the Schools for Health in Europe Rapid Assessment Tool” addresses a relevant and current theme, and presents an appropriate methodological design and data analysis.
Response: Thank you for your positive feedback. We appreciate your recognition of the relevance of the theme and the appropriateness of our methodological design and data analysis.
Overall, the manuscript presents a good level of scientific quality, but there are some points that require improvement.
Response: Thank you for your feedback. We have made efforts to address the points you raised and improve those aspects of the manuscript to enhance its overall quality. We appreciate your valuable suggestions.
In the introduction, more recent references should be added to the literature review to strengthen the argument. The argument regarding the relevance of the topic under study should also be developed.
Response: Thank you for your suggestion. We have further developed the argument regarding the relevance of the topic under study and we have added in the Introduction more recent references to the literature review[1–4] to strengthen the argument and provide a more comprehensive background.
It would be important to add information about the validation and limitations of the tool used for data collection.
Response: Thank you for your valuable comment. We have added information regarding the SHE rapid assessment tool, noting that it is recommended by the SHE network and supported by WHO Europe, being used across 43 member countries for planning, monitoring, and evaluation in schools. While the tool is widely applied in these contexts, there is no publicly available information on its reliability. Moreover, this study is the first to apply the tool within international schools, which underscores the novel contribution of our research in this area.
“In this study, nearly 60% of participants were women.” After this sentence it will not be necessary to add: “In contrast, men, comprising approximately 40% of the participants.” (line 250-251)
Response: Thank you for the comment. We have removed the sentence "In contrast, men, comprising approximately 40% of the participants" as suggested.
“This contrasts with other studies regarding with HPS, where only students aged between 8 and 18 years old were asked [20–22]” (line 255-256) It would be relevant to discuss the implications of this sample characteristic on the study results
Response: Thank you for the comment. We have addressed this point by discussing the implications of the broad age range: “The broad age range in this study, which includes students, teachers, and families, illustrates the diversity in age distribution among the respondents, indicating broad representation across different generations; this contrasts with other studies regarding HPSs, where only students aged between 8 and 18 years were included in the sample. This inclusive approach provides a more holistic view of health promotion within the school community, offering insights that may differ from those based solely on a student-focused sample”.
See also the impacts of some biases, of the profile of the participants, on the results obtained: about the fact that more women responded, about the response rate being 7%.
Response: Thank you for your comment. We have addressed the potential biases and the impact of participant profiles in the limitations section. We mention that the response rate was low (7%), and this, along with the overrepresentation of female respondents, could have influenced the results. Specifically, the feedback may be skewed due to participation being more likely from highly engaged individuals, potentially leading to an overestimation of the perceived success of health promotion initiatives. Despite this, we attempted to capture a diverse range of perspectives by including participants from various demographics. However, we acknowledge that the findings may not fully represent the entire school community.
“The notion of a HPS has proven effective in enhancing the health and well-being of students, while also aiding teaching and learning within educational settings.” (line 264-265) I don't understand the meaning of the sentence! It should be rewritten for better understanding.
Response: Thank you for your comment. We have rewritten the sentence to improve clarity and understanding of this phrase: “The concept of an HPS has been shown to effectively improve students` health and well-being of while also supporting their education and learning within educational settings”.
“The data from the SHE rapid assessment tool act as a starting point which will allow future planning for organizing to become a HPS [12, 16, 18].” (line 275-277) The sentence seems out of place to me!! It's not a discussion!
Response: Thank you for your comment. We included the sentence "The data from the SHE rapid assessment tool acts as a starting point which will allow future planning for organizing to become a HPS" in the discussion section to emphasize that the data serves as a foundation for future planning. The tool provides an initial assessment of health promotion efforts, and the insights gained from the data are intended to guide schools in their planning process to work towards becoming a Health Promoting School (HPS). This is why we positioned the statement in the discussion section to underline its practical application in future actions and improvements.
A review of the discussion is recommended in order to avoid repetition of ideas and improve the fluidity of the text. The comparison of results with other studies should be improved and deepened to reinforce the reading and interpretation of the results obtained.
Response: Thank you for your suggestion. We have reviewed the discussion to eliminate repetition and improve the fluidity of the text. Additionally, we have expanded and deepened the comparison of our results with other studies to better reinforce the interpretation of our findings. We believe these revisions strengthen the clarity and depth of the discussion.
Finally, it seems important to highlight more clearly the practical implications of the study carried out and suggest recommendations for future research.
Response: Thank you for your feedback. We have addressed this in the conclusions section, where we clearly highlight the practical implications of the study and provide recommendations for future research. We believe these additions offer a clearer direction for applying the study's findings and guide future investigations in this area.
I hope that my notes can contribute to the improvement of the presented manuscript.
Response: Thank you for your insightful comments. We truly appreciate your feedback, and we are confident that it has contributed to the improvement of the manuscript.
Comments on the Quality of English Language
It would be important to review the English used in this article to allow for a more fluid reading of the text.
Response: We appreciate your feedback. To ensure the highest quality and clarity in expressing the research, we have sent the text to American Journal Experts (AJE) for professional editing. We believe this will allow a more fluid reading of the text.
References
- Grupo de Trabajo de Escuelas Promotoras de Salud Guia Escuelas Promotoras de Salud; Madrid, 2023;
- OPS; OMS; UNESCO Hacer de Cada Escuela de La Región de Las Américas Una Escuela Promotora de Salud. Guía de Implementación Para Instituciones Educativas.; 2023;
- OPS; OMS; UNESCO Hacer Que Todas Las Escuelas Sean Promotoras de La Salud. Pautas e Indicadores Mundiales; 2022;
- Weber, M.W.; Black, M.; Carai, S.; Jullien, S. WHO Strategies to Improve Child and Adolescents Health in Europe. Global Pediatrics 2024, 9, 100215, doi:10.1016/j.gpeds.2024.100215.
Reviewer 2 Report
Comments and Suggestions for Authors
A review of the manuscript entitled “Health Promotion Evaluation in International Schools Using the Schools for Health in Europe Rapid Assessment Tool.”
- (line 17) The abstract states "nine International Schools", were these from different countries? Explain how schools were selected, whether participants were randomly recruited, and what statistical methods were used.
- (line 37) The authors did not explicitly state the research gap in their introduction. Why is this study needed?
- (line 58) There are other health assessment frameworks. Why is the SHE tool particularly useful for international schools?
- (line 90) The introduction outlines the study’s objective (assessing health promotion policies using the SHE tool), but no specific research questions or hypotheses are presented. Kindly provide research questions to clearly define the study’s scope and provide direction.
- (lines 40-41) This sentence “This phase of life is characterized by high receptivity, facilitating rapid acquisition of knowledge and habit formation.” will be more significant if it’s also supported by another relevant study. For example, https://pubmed.ncbi.nlm.nih.gov/39280290/
- (lines 96-98) The authors mentioned an estimated 12,000 individuals in the SEK Educational Community but did not clarify how many were invited to participate or how the final sample was selected.
- (lines 98-99) Were specific inclusion/exclusion criteria applied (e.g., only students aged 6–18, staff with a certain tenure)?
- (line 106) The authors did not mention ethical approval or informed consent in their methods, which are essential for studies involving human participants.
- (line 123) The survey used a 3-point scale for current status and priority, but why was this chosen instead of a 5- or 7-point Likert scale? A wider scale may have provided greater differentiation in responses.
- (line 220) While 245 participants provided open-ended feedback, there is no demographic breakdown. Who provided this feedback (students, staff, parents)? Did different groups express different concerns?
- (lines 192-198) The same idea was expressed twice:
"Regarding questions related to community links, all items were reported as either partly or totally in place in more than 75% of cases." (Line 192)
"Within questions related to community links all the items were partly or totally in place in more than 75% of cases." (Line 196)
- (lines 199-203) "However, in Question 7.3, ‘Our school promotes a healthy work-life balance, a reasonable workload, and provides an open environment to discuss work-related problems and stress,’ 35% of respondents believed it was not achieved." Who were these respondents—teachers, staff, or students? Did different stakeholder groups report different perspectives?
- (line 226) The main findings should be explicitly summarized at the beginning of the discussion. Right now, the discussion jumps into interpretation without a concise statement of the major findings.
- (line 360) The limitations section lacks depth and does not address key methodological concerns. Possible selection bias should be mentioned since only highly engaged participants may have responded.
- (lines 228-229) There is a misleading statement about representativeness: "The study offered valuable insights into various aspects of health promotion and its integration within an international school network, including a broad and diverse sample that proportionally represented different stakeholders in the educational community." Since the response rate was only 7%, making claims of proportional representation is inaccurate. The statement should be corrected as "The study provided insights into health promotion in an international school network based on a diverse sample. However, given the low response rate (7%), results may not fully represent the entire school community."
- (lines 310-315) I encourage authors to incorporate the study from Amirah et al. https://narrax.org/main/article/view/90/53 in the discussion, for example, "The positive feedback regarding the social environment, with school facilities promoting enjoyment and sociability, underscores the importance of creating supportive and inclusive environments for students. Evidence suggests that structured interventions, such as music therapy, can significantly enhance social interaction and communication skills, particularly in children with developmental challenges, such as Autism Spectrum Disorder (ASD) [47]. Implementing such approaches within school settings may further support social inclusion and student well-being."
- (line 379) The conclusion should begin with a brief summary of the study’s main findings. Currently, it jumps directly into the importance of the SHE rapid assessment tool without explicitly stating the study’s key results.
Comments on the Quality of English LanguageThe English language is good and easy to understand.
Author Response
Dear Reviewer 2,
Point-by-Point Response to Reviewer 2's comments:
Quality of English Language
The English could be improved to more clearly express the research.
Response: We appreciate your feedback. To ensure the highest quality and clarity in expressing the research, we have sent the text to American Journal Experts (AJE) for professional editing. This will help refine the language and ensure that the research is accurately and effectively communicated. Thank you for your suggestion.
Comments and Suggestions for Authors
A review of the manuscript entitled “Health Promotion Evaluation in International Schools Using the Schools for Health in Europe Rapid Assessment Tool.”
Response: Thank you for your detailed revision and valuable feedback. We really appreciate your time and insights.
- (line 17) The abstract states "nine International Schools", were these from different countries? Explain how schools were selected, whether participants were randomly recruited, and what statistical methods were used.
Response: The nine International Schools are part of a private network of schools. Participants were offered the opportunity to participate but were not randomly recruited, as we have explained in the limitations section. We used univariate analysis, and we have clarified this information in the abstract.
- (line 37) The authors did not explicitly state the research gap in their introduction. Why is this study needed?
Response: This study is necessary because the SHE rapid assessment tool is widely used to evaluate schools' health promotion policies and practices, as recommended by the SHE Network supported by WHO EURO. Despite its widespread use for needs assessments and situational analyses, no previous studies have analyzed the results obtained within a school community. This study fills that gap by evaluating the practical application of the tool in real-world school settings. We have stated this research gap in the Introduction section.
- (line 58) There are other health assessment frameworks. Why is the SHE tool particularly useful for international schools?
Response: We have clarified why this tool is particularly useful in the Introduction section.
The SHE (Schools for Health in Europe) rapid assessment tool is particularly useful for international schools because it provides a comprehensive, whole-school approach that considers not only the policies and practices but also the organizational, physical, and personal factors influencing health promotion. This is especially valuable in international schools, where the community may be diverse and dynamic. The tool allows schools to assess their current health and well-being-related practices, identify areas for improvement, and prioritize actions that are specific to their needs. Additionally, the tool can be used to track progress over time, providing an opportunity to reassess and adjust strategies as the school evolves.
Given its global recognition and support from WHO EURO, the SHE tool offers a trusted framework that aligns international schools with best practices in health promotion, making it an ideal resource for schools aiming to create a health-promoting environment that meets the diverse needs of their communities.
- (line 90) The introduction outlines the study’s objective (assessing health promotion policies using the SHE tool), but no specific research questions or hypotheses are presented. Kindly provide research questions to clearly define the study’s scope and provide direction.
Response: Thank you for your feedback. The questions that clearly define the study’s scope and provide direction are:
- What are the existing health promotion policies and practices in the selected international schools?
- How do these policies and practices align with global health promotion standards?
- What are the strengths and areas for improvement in these health promotion initiatives?
- How can health promotion efforts be enhanced to better meet the needs of the school community?
In response to your comment, we have outlined in the Introduction Section the research questions that clearly define the study’s scope and provide direction. “This research addresses the current health promotion policies and practices in a network of international schools, examining how they align with global health promotion standards. It explores the strengths and areas for improvement in these initiatives and identifies ways to enhance these efforts to better meet the needs of the school community”.
- (lines 40-41) This sentence “This phase of life is characterized by high receptivity, facilitating rapid acquisition of knowledge and habit formation.” will be more significant if it’s also supported by another relevant study. For example, https://pubmed.ncbi.nlm.nih.gov/39280290/
Response: Thank you for your suggestion. We have added the relevant study to support the statement. We appreciate your input.
- (lines 96-98) The authors mentioned an estimated 12,000 individuals in the SEK Educational Community but did not clarify how many were invited to participate or how the final sample was selected.
Response: Thank you for your comment. We would like to clarify that the estimated 12,000 individuals refer to an approximate census of the SEK Educational Community. This figure represents the total number of individuals within the community, but it is not the exact number of participants invited. As mentioned in the methods section, all potential participants received the survey link via internal institutional email. Additionally, parents could access the questionnaire through a link provided on the school’s online platform. We have added that this ensures all members of the community had the opportunity to participate.
- (lines 98-99) Were specific inclusion/exclusion criteria applied (e.g., only students aged 6–18, staff with a certain tenure)?
Response: Thank you for your question. We have specified that the survey targeted students between 6 and 18 years old, teaching and non-teaching staff, as well as families within the SEK Educational Community who were able to understand the questions. There were no specific exclusion criteria beyond this, as the survey was designed to gather insights from a broad range of community members.
- (line 106) The authors did not mention ethical approval or informed consent in their methods, which are essential for studies involving human participants.
Response: Thank you for your comment. The ethical approval and informed consent details are addressed in both the "Institutional Review Board Statement" and the "Informed Consent Statement" sections. The study was conducted in accordance with the Declaration of Helsinki and approved by the Research Ethics Committee of UCJC (approval code: 1121, approval date: 01/03/2019). All applicable data protection laws were strictly adhered to.
Informed consent was obtained from all participants, who were fully informed about the study’s purpose, their voluntary participation, and the use of their data. Participants consented to completing the questionnaire and allowing their data to be analyzed for research purposes, ensuring compliance with ethical guidelines and maintaining confidentiality.
- (line 123) The survey used a 3-point scale for current status and priority, but why was this chosen instead of a 5- or 7-point Likert scale? A wider scale may have provided greater differentiation in responses.
Response: Thank you for your comment. The 3-point scale was chosen because the SHE rapid assessment tool, designed by the Schools for Health in Europe (SHE) network, specifically employs this approach for assessing both current status and priority.
- Current status: This evaluates the school’s current situation on a three-point scale (1 = not in place, 2 = partly in place, 3 = fully in place).
- Priority: This assesses the priority the school community gives to the issue, using a three-point scale (1 = low/no priority, 2 = medium priority, 3 = high priority).
Response: We have explained in the Methods Section that this 3-point scale ensures clarity and simplicity, making it easier for schools to quickly assess and identify areas for improvement without unnecessary complexity. The scale aligns with the tool’s purpose of providing a rapid yet effective evaluation of health promotion initiatives.
- (line 220) While 245 participants provided open-ended feedback, there is no demographic breakdown. Who provided this feedback (students, staff, parents)? Did different groups express different concerns?
Response: The open-ended feedback was provided by different members of the school community, including students, staff, and parents. As the study examined the entire school community as a whole, the feedback reflects a collective perspective. The study was not specifically aimed at analyzing differences by school groups, so the responses were not differentiated by demographic categories. However, the feedback provides a broad overview of the health promotion initiatives from the perspective of the entire community, as explained in the Methods section.
- (lines 192-198) The same idea was expressed twice: "Regarding questions related to community links, all items were reported as either partly or totally in place in more than 75% of cases." (Line 192) "Within questions related to community links all the items were partly or totally in place in more than 75% of cases." (Line 196)
Response: Thank you for your query. We have eliminated the text in line 196 to express this idea only once.
- (lines 199-203) "However, in Question 7.3, ‘Our school promotes a healthy work-life balance, a reasonable workload, and provides an open environment to discuss work-related problems and stress,’ 35% of respondents believed it was not achieved." Who were these respondents—teachers, staff, or students? Did different stakeholder groups report different perspectives?
Response: Thank you for your comment. As explained earlier, the study examined the entire school community as a whole, so the feedback reflects a collective perspective. The study was not specifically aimed at analyzing differences by school groups, so the responses were not differentiated by demographic categories. However, the survey provides a broad overview of the health promotion initiatives from the perspective of the entire community, as outlined in the Methods section.
- (line 226) The main findings should be explicitly summarized at the beginning of the discussion. Right now, the discussion jumps into interpretation without a concise statement of the major findings.
Response: Thank you for your suggestion. We have updated the 'Main Findings' section of the discussion to explicitly summarize the key findings at the beginning. This revision ensures a clear and concise statement of the major findings before diving into interpretation, addressing the concern of jumping directly into the analysis without first summarizing the results.
- (line 360) The limitations section lacks depth and does not address key methodological concerns. Possible selection bias should be mentioned since only highly engaged participants may have responded.
Response: Thank you for your comment. In addition to the methodological concerns already mentioned in the limitations section (such as the descriptive nature of the study, response bias, and non-probabilistic sampling), we have also addressed the possibility of selection bias. The survey may have been disproportionately completed by highly engaged participants, potentially leading to an overestimation of the perceived success of health promotion initiatives. This addition enhances the depth of the limitations and more comprehensively addresses key methodological concerns. However, despite these limitations, we have emphasized that the methodology provided a valuable opportunity for an in-depth exploration of health promotion challenges in educational settings.
- (lines 228-229) There is a misleading statement about representativeness: "The study offered valuable insights into various aspects of health promotion and its integration within an international school network, including a broad and diverse sample that proportionally represented different stakeholders in the educational community." Since the response rate was only 7%, making claims of proportional representation is inaccurate. The statement should be corrected as "The study provided insights into health promotion in an international school network based on a diverse sample. However, given the low response rate (7%), results may not fully represent the entire school community."
Response: Thank you for this clarification. We have corrected the statement as requested: “This study offers valuable insights into health promotion in an international school network based on a diverse sample. However, given the low response rate (7%), the results may not fully represent the entire school community".
- (lines 310-315) I encourage authors to incorporate the study from Amirah et al. https://narrax.org/main/article/view/90/53 in the discussion, for example, "The positive feedback regarding the social environment, with school facilities promoting enjoyment and sociability, underscores the importance of creating supportive and inclusive environments for students. Evidence suggests that structured interventions, such as music therapy, can significantly enhance social interaction and communication skills, particularly in children with developmental challenges, such as Autism Spectrum Disorder (ASD). Implementing such approaches within school settings may further support social inclusion and student well-being."
Response: Thank you for your suggestion. We have incorporated the study by Amirah et al. as recommended and have added the suggested text to the discussion. This addition strengthens our argument by further emphasizing the importance of structured interventions, such as music therapy, in fostering social inclusion and student well-being within school settings
- (line 379) The conclusion should begin with a brief summary of the study’s main findings. Currently, it jumps directly into the importance of the SHE rapid assessment tool without explicitly stating the study’s key results.
Response: Thank you for your suggestion. We have revised the conclusion to begin with a brief summary of the study’s main findings before discussing the importance of the SHE rapid assessment tool. The revised conclusion now acknowledges the significant efforts made by international schools in integrating health promotion initiatives, the prioritization of health-related issues by stakeholders, and the identification of areas for improvement, such as staff well-being and access to physical activity. These additions ensure that the conclusion provides a more comprehensive synthesis of the study while maintaining a clear connection to the findings.
Comments on the Quality of English Language: The English language is good and easy to understand.
Response: We appreciate you think the English language is good and easy to understand. However, to ensure the highest quality and clarity in expressing the research, we have sent the text to American Journal Experts (AJE) for professional editing. This will help refine the language and ensure that the research is accurately and effectively communicated.
Reviewer 3 Report
Comments and Suggestions for Authors
The manuscript presents a study evaluating health promotion policies and practices in international schools using the SHE Rapid Assessment Tool. While it contributes to an important field, there are notable concerns regarding novelty, methodology, structuring, writing quality, and citation accuracy.
The study focuses on an underexplored context: international schools.
The study heavily relies on existing frameworks (SHE network) without significant new insights beyond the application of a known tool.
The findings are predictable based on existing literature; it does not introduce a new theoretical model or framework.
The objectives are clearly defined. However, the objectives focus on descriptive evaluation rather than generating statistical relationships. Clearly state whether the study aims for exploratory or evaluative outcomes. The manuscript also lacks hypothesis or research questions. Consider statistical comparisons (e.g., chi-square, t-tests) to strengthen findings.
The use of a validated tool (SHE) is appropriate and there is a diverse sample (students, teachers, families, etc.). Please provide the full questionnaire (37 questions of SHE) into the main text or Appendix.
Sampling bias: The response rate of 7% is extremely low, raising external validity concerns. Please acknowledge the low response rate as a limitation.
No control group: It is unclear how the results compare to other health assessment models.
Data analysis lacks rigor: There is no mention of statistical tests for significance, only descriptive statistics.
Figures 1-4 effectively summarize data. The findings do not indicate whether observed differences are statistically significant. Use inferential statistics to determine significant differences.
Findings not sufficiently explained: Example: "30% of respondents were dissatisfied with health assessment policies". What factors contributed to this dissatisfaction?
Redundancy: Some result descriptions (especially the social environment and health skills sections) repeat information. Summarize redundant content more concisely.
Policy recommendations lack specificity: The discussion mentions improving stakeholder engagement but does not propose concrete strategies. No discussion of feasibility: Implementing SHE-based programs requires resources—how feasible is this for underfunded international schools? Provide specific implementation strategies, especially for schools with limited resources.
Line 39: "Education attendance is mandatory and accessible for individuals aged 4 to 16 years [3]."Clarify regional context (Is this in Spain, EU, or globally?)
Line 52: "ENHPS... began with the participation of seven countries [12]." Consider listing these countries for specificity.
Line 96: "A descriptive cross-sectional study with an analytical approach..." Clarify what "analytical approach" entails.
Line 110: "All potential participants received the survey link via email..." Potential bias (Were reminders sent? How was response encouraged?)
Line 157: "A total of 929 participants..." Add response rate (7%) here for transparency.
Line 227: "The study offered valuable insights..." Clarify what is 'valuable' beyond existing knowledge.
Line 349: "Findings underscore the importance of prioritizing health promotion..." Avoid generic phrasing. What is the most significant policy takeaway?
Author Response
Dear Reviewer 3,
Point-by-Point Response to Reviewer 3's comments:
Quality of English Language
The English is fine and does not require any improvement.
Response: We appreciate you think the English is fine and does not require any improvement. However, to ensure the highest quality and clarity in expressing the research, we have sent the text to American Journal Experts (AJE) for professional editing. This will help refine the language and ensure that the research is accurately and effectively communicated.
Comments and Suggestions for Authors
The manuscript presents a study evaluating health promotion policies and practices in international schools using the SHE Rapid Assessment Tool. While it contributes to an important field, there are notable concerns regarding novelty, methodology, structuring, writing quality, and citation accuracy.
Response: We respectfully disagree with the reviewer’s assessment. The SHE Rapid Assessment Tool has never been explored for research purposes, making our study highly novel. Additionally, we have made significant efforts to improve the methodology, structure, writing quality, and citation accuracy to ensure clarity and rigor. We have implemented several changes throughout the text to enhance these aspects and provide a more precise and well-structured analysis.
The study focuses on an underexplored context: international schools.
Response: We have not found any studies examining Health Promoting Schools and the use of this tool for improvement, which is why we do not share the reviewer’s opinion. This topic remains unexplored and deserves further investigation. Our study contributes to filling this gap by providing valuable insights into health promotion in international schools using the SHE Rapid Assessment Tool, as detailed in the manuscript.
The study heavily relies on existing frameworks (SHE network) without significant new insights beyond the application of a known tool.
Response: Thank you for your feedback. While the SHE rapid assessment tool is an established framework, this study offers new insights by analyzing data from a network of schools, something that has not been done before. The tool has traditionally been used by individual schools for planning purposes, but this study aggregates responses across multiple schools, revealing trends, gaps, and priorities that are not visible when examining schools in isolation. By applying the tool at a network level, we provide novel insights into the effectiveness of health promotion practices across a broader context, offering actionable findings that extend beyond individual schools. This approach contributes new knowledge to the field, demonstrating the tool's impact in a way that has not been explored previously.
The findings are predictable based on existing literature; it does not introduce a new theoretical model or framework.
Response: We respectfully disagree with this assessment. While our study does not introduce a new theoretical model, it provides a research-based evaluation of the SHE rapid assessment tool in a way that has not been done before. This study moves beyond theoretical discussions by demonstrating the tool’s practical application across a network of schools, generating empirical data that can be compared with existing literature. By systematically analyzing how schools use the tool and the insights it produces, we offer new evidence on its effectiveness and applicability. This contributes valuable, research-driven findings that enhance understanding of health promotion in schools, complementing existing theoretical frameworks with practical, data-based insights.
The objectives are clearly defined. However, the objectives focus on descriptive evaluation rather than generating statistical relationships. Clearly state whether the study aims for exploratory or evaluative outcomes. The manuscript also lacks hypothesis or research questions. Consider statistical comparisons (e.g., chi-square, t-tests) to strengthen findings.
Response: Thank you for your feedback. The questions that define the study’s scope and provide direction are:
- What are the existing health promotion policies and practices in the selected international schools?
- How do these policies and practices align with global health promotion standards?
- What are the strengths and areas for improvement in these health promotion initiatives?
- How can health promotion efforts be enhanced to better meet the needs of the school community?
In response to your comment, we have outlined in the Introduction Section the research questions that clearly define the study’s scope and provide direction. “This research addresses the current health promotion policies and practices in a network of international schools, examining how they align with global health promotion standards. It explores the strengths and areas for improvement in these initiatives and identifies ways to enhance these efforts to better meet the needs of the school community”.
This study is intentionally descriptive, as it analyzes the school community as a whole rather than focusing on bivariate or multivariable differences. The objective is to evaluate the practical application of the SHE rapid assessment tool rather than to establish statistical relationships. However, we did conduct a statistical univariate analysis of independent and dependent variables to provide further insight into the data. While we recognize the potential value of statistical comparisons (e.g., chi-square, t-tests), this would be better suited as an objective for future research in this line of study.
The use of a validated tool (SHE) is appropriate and there is a diverse sample (students, teachers, families, etc.). Please provide the full questionnaire (37 questions of SHE) into the main text or Appendix.
Response: Thank you for your feedback. We appreciate the recognition of the appropriateness of the validated SHE tool and the diversity of our sample. As requested, the full questionnaire (37 questions of the SHE rapid assessment tool) is included in Supplementary File S1 “Schools for Health in Europe (SHE) Rapid Assessment Tool Questions”.
Sampling bias: The response rate of 7% is extremely low, raising external validity concerns. Please acknowledge the low response rate as a limitation.
Response: Thank you for your feedback. We acknowledge the concern regarding the low response rate. As noted in the limitations section, we explicitly state: "However, given the low response rate (7%), the results may not fully represent the entire school community."
This acknowledges the potential impact on external validity while ensuring transparency about the study’s limitations.
No control group: It is unclear how the results compare to other health assessment models.
Response: Thank you for your feedback. As this is a descriptive study, its primary aim is to assess the practical application of the SHE rapid assessment tool rather than to establish causality or temporality. We have acknowledged these limitations in the manuscript and recognize that future comparative or longitudinal studies could further strengthen insights in this research area.
Data analysis lacks rigor: There is no mention of statistical tests for significance, only descriptive statistics.
Response: As we have explained, this study is intentionally descriptive, as it analyzes the school community as a whole rather than focusing on bivariate or multivariable differences. The objective is to evaluate the practical application of the SHE rapid assessment tool rather than to establish statistical relationships.
However, we have conducted a statistical univariate analysis of independent and dependent variables to provide further insight into the data. While we recognize the potential value of statistical comparisons (e.g., chi-square, t-tests), this would be better suited as an objective for future research in this line of study.
Figures 1-4 effectively summarize data. The findings do not indicate whether observed differences are statistically significant. Use inferential statistics to determine significant differences.
Response: Thank you for your feedback. We do not intend to use inferential statistics to determine significant differences, as the goal of this study is purely descriptive. Our focus is on summarizing and describing the results of the analysis of the SHE rapid assessment tool, rather than making statistical inferences about significant differences. The figures (1-4) effectively summarize the data to provide an overview of the findings, and the objective is to evaluate the practical application of the tool within the context of the school community.
Findings not sufficiently explained: Example: "30% of respondents were dissatisfied with health assessment policies". What factors contributed to this dissatisfaction?
Response: Thank you for this comment. We acknowledge the importance of understanding the factors contributing to the dissatisfaction with health assessment policies. However, the SHE rapid assessment tool does not include questions that specifically identify the factors behind the responses on the 3-point scale. As such, we do not have this information in the current study. We recognize that exploring these underlying factors could be valuable, and this could serve as an objective for future lines of research.
Redundancy: Some result descriptions (especially the social environment and health skills sections) repeat information. Summarize redundant content more concisely.
Response: We appreciate your observation regarding redundancy in the result descriptions, particularly in the social environment and health skills sections. In response, we have carefully revised the manuscript and removed the repeated information to present the findings more concisely.
Policy recommendations lack specificity: The discussion mentions improving stakeholder engagement but does not propose concrete strategies. No discussion of feasibility: Implementing SHE-based programs requires resources—how feasible is this for underfunded international schools? Provide specific implementation strategies, especially for schools with limited resources.
Response: Thank you for your thoughtful feedback. We agree that the policy recommendations should be more specific, and we appreciate your suggestion for concrete strategies. To improve stakeholder engagement, one actionable strategy could be organizing regular, short meetings or workshops with teachers, students, and families to raise awareness about the importance of health promotion. These can be low-cost initiatives, such as virtual meetings or utilizing existing school events to discuss health policies. Another approach could be implementing a “Health Ambassador” program, where selected students or teachers are tasked with promoting health initiatives within the school community. This fosters ownership and reduces the reliance on external resources.
In terms of feasibility, particularly for underfunded international schools, an actionable strategy could be starting small by integrating health promotion into existing school activities. For instance, incorporating health-related topics into the curriculum or school assemblies requires minimal financial investment but can still have a significant impact. Additionally, schools can collaborate with local health organizations or universities to secure pro bono resources or expert advice, which would help implement SHE-based programs without incurring high costs. These strategies make it possible to implement health promotion programs in schools with limited resources, leveraging existing assets and forming partnerships to achieve meaningful results.
Line 39: "Education attendance is mandatory and accessible for individuals aged 4 to 16 years [3]."Clarify regional context (Is this in Spain, EU, or globally?)
Response: Thank you for your request for clarification. In response, we have updated the manuscript to specify the regional context. The sentence now reads: " Education attendance is mandatory and accessible for individuals aged 4 to 16 years in many regions across the globe". This provides clearer information while maintaining the intended global perspective.
Line 52: "ENHPS... began with the participation of seven countries [12]." Consider listing these countries for specificity.
Response: Thank you for your suggestion. We appreciate the importance of specifying the countries involved in the early stages of the European Network of Health Promoting Schools (ENHPS). The seven countries that participated in the initial phase were: Austria, Belgium, Finland, Germany, Netherlands, Portugal, and Sweden. We have updated the manuscript to include this information for greater specificity.
Line 96: "A descriptive cross-sectional study with an analytical approach..." Clarify what "analytical approach" entails.
Response: Thank you for your comment. To clarify, while this is a descriptive cross-sectional study, we aimed to provide an analysis of the responses collected through the SHE rapid assessment tool. While the primary focus was on describing the current state of health promotion in the school communities, we also analyzed the data to identify key patterns and insights. This analytical approach involved examining the relationships between different responses to gain a deeper understanding of the data, even though we did not perform inferential statistical tests. We have updated the manuscript to further clarify this distinction.
Line 110: "All potential participants received the survey link via email..." Potential bias (Were reminders sent? How was response encouraged?)
Response: Thank you for your clarification. To address the concern, the potential bias in our study stems from the fact that only individuals who were already interested in the topic may have chosen to participate, as we did not send reminders. The encouragement from school nurses likely helped engage some participants, but it is possible that those with a stronger interest in health promotion were more inclined to respond. We have updated the manuscript to reflect this and have acknowledged this limitation in the discussion.
Line 157: "A total of 929 participants..." Add response rate (7%) here for transparency.
Response: Thank you for your suggestion. We have added the response rate of 7% to the manuscript for transparency
Line 227: "The study offered valuable insights..." Clarify what is 'valuable' beyond existing knowledge.
Response: Thank you for your comment. To clarify, the term "valuable" refers to the unique application of the SHE rapid assessment tool in a research context, as it has primarily been used for planning purposes rather than evaluation. By using this tool in an actual research study, we gained insights into how the tool performs in assessing health promotion within school communities, which had not been explored before. Additionally, the study highlights areas of strength and opportunities for improvement in schools' health promotion efforts, which can inform future actions and research in this field. We have updated the manuscript to specify these contributions to the existing knowledge.
Line 349: "Findings underscore the importance of prioritizing health promotion..." Avoid generic phrasing. What is the most significant policy takeaway?
Response: Thank you for your comment. To avoid generic phrasing, we have revised the section to clearly highlight the most significant policy takeaway from the study. The key takeaway is that schools should prioritize the integration of health promotion into their core policies and practices, with a specific focus on enhancing stakeholder engagement. This includes involving teachers, students, and families more actively in shaping and implementing health promotion initiatives. Strengthening these collaborative efforts will be crucial for creating sustainable health-promoting environments within schools. We have updated the manuscript to reflect this more specific policy recommendation.
Reviewer 4 Report
Comments and Suggestions for Authors
Dear Authors,
Congratulations on your work. The following comments provide a detailed evaluation of the study, highlighting areas for improvement and suggesting ways to enhance clarity, coherence, and analytical depth.
Title: Health Promotion Evaluation in International Schools Using the Schools for Health in Europe Rapid Assessment Tool
Abstract
- The study's context should be briefly mentioned, indicating that it has been applied in several countries.
- The type of survey conducted should be specified in the abstract.
- The characterization of the sample is not a direct result of the study’s objective.
- The abstract does not highlight the diversity of nationalities (74) or the full age range (7–77 years), both of which are important findings.
- The abstract should briefly highlight any international context findings or challenges to justify the specificity in the title.
Introduction
- The introduction extensively discusses the background and history of the Schools for Health in Europe (SHE) network but does not clearly introduce the specific study context (international schools) or explain why this population was chosen. This clarification is found in the Methods section (lines 95–105), where it states that the study was conducted in SEK international schools, but this should be explicitly mentioned in the introduction.
- The introduction should justify why the SHE rapid assessment tool is the most appropriate method for evaluating health promotion policies in international schools.
Methodology
- Ethical approval should be explicitly stated in the methodology rather than being placed at the end of the article. This information should be moved or duplicated in the Methods section, possibly under Study Design or procedures (lines 95–105).
- The methodology states that data collection occurred via an online survey (line 107) but does not specify which platform was used or how response bias was mitigated.
- The methodology mentions the use of the SHE rapid assessment tool (line 117) but does not provide any information regarding its reliability (e.g., Cronbach’s alpha) or whether it has been validated for use in international schools.
Results
- The characterization of the sample is not a direct result of the study’s objective and would be better placed in the sampling section.
- The abstract states that 30% of respondents expressed dissatisfaction with how schools assess health needs and promote work-life balance (lines 24–25). However, the results section states, “Respondents were asked whether the school has evaluated the needs and desires of students and staff regarding health and well-being. Over a third indicated that this aspect was not currently achieved.” These statements should be aligned for consistency.
- The results report only percentages and means but do not compare responses across groups (e.g., students vs. teachers, age groups). This contradicts the analytical approach mentioned in the methodology (line 96). If inferential analyses (e.g., chi-square tests, t-tests) were not conducted, the authors should justify this limitation and mention it in the discussion.
- The title and methodology suggest a comparison across different international schools and countries, yet the results section does not provide any comparative analysis. Instead, data is aggregated from all schools without distinguishing between locations, which creates a mismatch in expectations.
- The figure 4 in the results is not explained in text form like the other figures. A small explanation should enhance reader comprehension.
Discussion
- The discussion repeatedly states that schools are making progress in health promotion (lines 231–236, 278–279) but does not critically evaluate which areas require the most improvement based on the results. However, the abstract and results section highlight that 30% of respondents were dissatisfied with how the school assesses health needs and 35% felt that work-life balance was not supported (lines 24–25, 200–202). These challenges should be more explicitly emphasized in the discussion.
- The study includes schools from multiple countries (Spain, France, Qatar, Ireland) (lines 99–104), but the discussion does not compare findings across schools, countries, or cultural contexts, despite mentioning the presence of 74 nationalities (lines 257–259).
- The discussion states that most elements of health promotion are in place, but the results indicate that key areas like work-life balance and health needs assessment have significant gaps. These negative findings should be addressed more explicitly rather than focusing primarily on positive results.
- The discussion highlights gaps in stakeholder participation and physical activity policies (lines 297–307) but does not suggest specific actions for schools. What specific strategies should schools adopt based on these findings?
Author Response
Dear Reviewer 4,
Point-by-Point Response to Reviewer 4's comments:
The English is fine and does not require any improvement.
Response: We appreciate you think the English is fine and does not require any improvement. However, to ensure the highest quality and clarity in expressing the research, we have sent the text to American Journal Experts (AJE) for professional editing. This will help refine the language and ensure that the research is accurately and effectively communicated.
Comments and Suggestions for Authors
Dear Authors,
Congratulations on your work. The following comments provide a detailed evaluation of the study, highlighting areas for improvement and suggesting ways to enhance clarity, coherence, and analytical depth.
Response: Thank you for your detailed evaluation and constructive feedback. We appreciate the time and effort you have put into reviewing the study. We have tried to improve clarity, coherence and analytical depth based on your suggestions.
Abstract
- The study's context should be briefly mentioned, indicating that it has been applied in several countries.
Response: Thank you for your suggestion. We have revised the abstract to briefly mention the study's context, emphasizing that it has been applied in multiple countries.
- The type of survey conducted should be specified in the abstract.
Response: We have specified in the abstract that a cross-sectional survey was conducted based on the SHE rapid assessment tool, which comprises a series of 37 questions aligned with the whole-school approach to health promotion. These questions address orientation, healthy school policy, physical environment, social environment, health skills, community links, and the healthy school team, using a 3-point scale
- The characterization of the sample is not a direct result of the study’s objective.
Response: While the characterization of the sample is not the primary objective of the study, it provides essential context for interpreting the results. Understanding the distribution of participants across different groups within the school community allows for a more nuanced analysis of varying perspectives on health promotion policies and practices. As stated in the Variables subsection, these characteristics are among the variables analyzed and including them in the abstract ensures a clearer representation of the study population. We hope this clarifies the rationale for their placement.
- The abstract does not highlight the diversity of nationalities (74) or the full age range (7–77 years), both of which are important findings.
Response: Thank you for your feedback. We have updated the abstract to highlight the diversity of nationalities (74) and the full age range (7–77 years), ensuring that these important findings are clearly presented.
- The abstract should briefly highlight any international context findings or challenges to justify the specificity in the title.
Response: We have introduced a brief mention of the international context findings and challenges in the abstract to better justify the specificity in the title.
Introduction
- The introduction extensively discusses the background and history of the Schools for Health in Europe (SHE) network but does not clearly introduce the specific study context (international schools) or explain why this population was chosen. This clarification is found in the Methods section (lines 95–105), where it states that the study was conducted in SEK international schools, but this should be explicitly mentioned in the introduction.
Response: Thank you for your feedback. We have revised the introduction to explicitly mention the focus on SEK International Schools and the rationale for selecting this population: “The main objective was to assess the current health promotion policies and practices within the SEK International School network via the SHE rapid assessment tool. The rationale for conducting this research within this school network is the collaboration between SEK International Schools and the master’s degree program in school health at Camilo José Cela University.”
- The introduction should justify why the SHE rapid assessment tool is the most appropriate method for evaluating health promotion policies in international schools.
Response: Thank you for your feedback. As we have previously noted, the introduction now explicitly justifies the use of the SHE rapid assessment tool, which is the recommended method for evaluating health promotion policies by the SHE network and is supported by WHO Europe. This study goes beyond theoretical discussions by providing empirical data on the tool’s practical application across a network of international schools, something that has not been done before.
By aggregating responses from multiple schools, we reveal trends and gaps that individual school assessments cannot uncover. This approach fills a gap in the literature, demonstrating the tool’s effectiveness in real-world settings and offering actionable insights that extend beyond individual school
Methodology
- Ethical approval should be explicitly stated in the methodology rather than being placed at the end of the article. This information should be moved or duplicated in the Methods section, possibly under Study Design or procedures (lines 95–105).
Response: Thank you for your comment. We have placed the ethical approval information at the end of the manuscript because the journal’s guidelines specify that the Institutional Review Board and Informed Consent statements should be included in that section.
- The methodology states that data collection occurred via an online survey (line 107) but does not specify which platform was used or how response bias was mitigated.
Response: Thank you for your comment. In the Methods section, we explained that potential participants received the survey link via internal institutional email, and parents could access the questionnaire through a link on the school’s online platform, which is designed to keep families informed about their children's school life. This ensured that all members of the school community had the opportunity to participate, with support from school nurses and other staff.
We also acknowledge response bias as a limitation in the Limitations section of the manuscript, where we discuss potential biases in participation. We hope this clarification addresses your concern.
- The methodology mentions the use of the SHE rapid assessment tool (line 117) but does not provide any information regarding its reliability (e.g., Cronbach’s alpha) or whether it has been validated for use in international schools.
Response: The SHE rapid assessment tool is recommended by the SHE network and supported by WHO Europe, being used across 43 member countries for planning, monitoring, and evaluation in schools. While the tool is widely applied in these contexts, there is no publicly available information on its reliability. Moreover, this study is the first to apply the tool within international schools for research purposes, which underscores the novel contribution of our research in this area.
Results
- The characterization of the sample is not a direct result of the study’s objective and would be better placed in the sampling section.
Response: Thank you for your comment. We respectfully disagree with the suggestion to move the characterization of the sample to the sampling section. The sociodemographic variables are integral to the results of the study, as they provide essential context for interpreting the findings. As stated in the Variables subsection, these characteristics are among the variables analyzed and including them in the abstract and results section ensures a clearer understanding of the study population. We hope this clarifies the reasoning behind their placement.
- The abstract states that 30% of respondents expressed dissatisfaction with how schools assess health needs and promote work-life balance (lines 24–25). However, the results section states, “Respondents were asked whether the school has evaluated the needs and desires of students and staff regarding health and well-being. Over a third indicated that this aspect was not currently achieved.” These statements should be aligned for consistency.
Response: We have revised the abstract to state, "30% of respondents indicated that the school had not yet achieved the assessment of students' and employees' health and well-being needs or effectively promoted a healthy work-life balance". This aligns with the results section, which states, "Respondents were asked whether the school has evaluated the needs and desires of students and staff regarding health and well-being. Over a third indicated that this aspect was not currently achieved".
- The results report only percentages and means but do not compare responses across groups (e.g., students vs. teachers, age groups). This contradicts the analytical approach mentioned in the methodology (line 96). If inferential analyses (e.g., chi-square tests, t-tests) were not conducted, the authors should justify this limitation and mention it in the discussion.
Response: Thank you for your comment. To clarify, this is a descriptive cross-sectional study aimed at analyzing responses collected through the SHE rapid assessment tool. While the primary focus was to describe the current state of health promotion in the school communities, we also examined patterns and insights within the data. However, inferential statistical tests (e.g., chi-square, t-tests) were not conducted, as the study's objective was not to establish statistical associations but rather to provide an overview of health promotion practices. We have updated the manuscript to clarify this distinction, and we have acknowledged this as a limitation in the discussion
- The title and methodology suggest a comparison across different international schools and countries, yet the results section does not provide any comparative analysis. Instead, data is aggregated from all schools without distinguishing between locations, which creates a mismatch in expectations.
Response: Thank you for your comment. As explained earlier, the study aimed to evaluate health promotion within the entire school community as a whole, rather than conducting a comparative analysis between different schools or countries. The schools in this network operate under the same international framework, and the objective was to assess overall health promotion efforts rather than differences across locations. This approach is clearly outlined in the Methods section. While data was aggregated to reflect a collective perspective, future studies could explore potential variations between individual schools or regions.
- The figure 4 in the results is not explained in text form like the other figures. A small explanation should enhance reader comprehension.
Response: Thank you for the suggestion. We have added a brief explanation in the results section to enhance reader comprehension: "Regarding the 'current (in place)' status, responses generally fall between 'partly' and 'fully' implemented. On the other hand, for 'priority,' most responses indicate 'high' or 'medium' importance across all seven spheres”.
Discussion
- The discussion repeatedly states that schools are making progress in health promotion (lines 231–236, 278–279) but does not critically evaluate which areas require the most improvement based on the results. However, the abstract and results section highlight that 30% of respondents were dissatisfied with how the school assesses health needs and 35% felt that work-life balance was not supported (lines 24–25, 200–202). These challenges should be more explicitly emphasized in the discussion.
Response: We have revised the discussion to explicitly emphasize the key challenges identified in the results: "While most goals for supporting healthy school staff were met, more than a third of respondents felt that promoting work-life balance and managing stress was not adequately achieved, despite being identified as a priority. This highlights a critical gap in staff well-being initiatives. Other studies also emphasize the urgent need for schools to prioritize comprehensive strategies that address staff health, reduce workplace stress, and foster a more supportive and positive working environment." We have incorporated this section to directly address the reviewer's comment.
- The study includes schools from multiple countries (Spain, France, Qatar, Ireland) (lines 99–104), but the discussion does not compare findings across schools, countries, or cultural contexts, despite mentioning the presence of 74 nationalities (lines 257–259).
Response: As we have explained, this study aimed to evaluate health promotion within the entire school community as a whole, rather than conducting a comparative analysis between different schools or countries. The participating schools operate under the same international framework, and the objective was to assess overall health promotion efforts rather than regional differences. This approach is clearly outlined in the Methods section. While data was aggregated to provide a collective perspective, future research could explore potential variations between individual schools or cultural contexts to further expand on these findings.
- The discussion states that most elements of health promotion are in place, but the results indicate that key areas like work-life balance and health needs assessment have significant gaps. These negative findings should be addressed more explicitly rather than focusing primarily on positive results.
Response: Thank you for your comment. We have addressed these negative findings more explicitly in the discussion: "While most goals for supporting healthy school staff were met, more than a third of respondents felt that promoting work-life balance and managing stress was not adequately achieved, despite being identified as a priority. This highlights a critical gap in staff well-being initiatives. Other studies also emphasize the urgent need for schools to prioritize comprehensive strategies that address staff health, reduce workplace stress, and foster a more supportive and positive working environment." This revision ensures a balanced discussion by acknowledging both strengths and areas needing improvement.
- The discussion highlights gaps in stakeholder participation and physical activity policies (lines 297–307) but does not suggest specific actions for schools. What specific strategies should schools adopt based on these findings?
Response: Thank you for your suggestion. We have included strategies such as “conducting regular surveys to gather feedback from all stakeholders, organizing focus groups to ensure a deeper understanding of varying perspectives, and creating opportunities for open discussions about the importance of health promotion activities. Additionally, schools could consider providing tailored professional development for staff to align their views on health priorities, while actively involving students and parents in decision-making processes to ensure a more comprehensive and inclusive approach to health promotion.
Round 2
Reviewer 2 Report
Comments and Suggestions for Authors
I commend the authors for addressing my concerns on version 1 of their manuscript. However, I notice that they haven’t fully addressed my comments, specifically on lines 46-47, where they stated that “This phase of life is characterised by high receptivity, facilitating rapid acquisition of knowledge and habit formation.” This statement will be more significant if it’s also supported by another relevant study. Kindly cite https://pubmed.ncbi.nlm.nih.gov/39280290/
(page 1, line 24) Did the authors really include children starting at age 7 in their survey? If yes, then my concern is that their answers may not be reliable due to limited cognitive development, memory, and reasoning skills. Did authors provide those children with surveys that are child-friendly, such as simple questions?
Author Response
Dear reviewer 2,
We sincerely appreciate your new review and valuable comments on our manuscript. We are also grateful for your positive feedback on the English language, as it was revised by native professional speakers.
Additionally, we are pleased that the research design, results, and conclusions now appear appropriate to you.
To improve the introduction, we have incorporated the requested citation, which had already been referenced in a previous version of the manuscript. However, due to an issue with the reference manager, it was not properly updated in the bibliography. This has now been corrected.
Regarding the participation of five 7-year-old children in the survey, as explained in the methods section, the questionnaire was offered to the entire educational community. Only five children of this age chose to respond, as they considered themselves capable of understanding the questions. To clarify any possible doubts, they received support from school nurses, who helped explain the questions. Additionally, respondents had the option to select "NA/DN" (No Answer/Don’t Know) if they did not understand a question. We hope this explanation further reinforces the clarifications provided in the methods section.
Once again, we would like to emphasize that the purpose of this SHE network rapid assessment tool is to evaluate the state of health promotion in schools based on priority and implementation status. Therefore, any response provides relevant information for assessing health promotion activities in schools, allowing for the planning and implementation of actions to improve areas requiring further development.
Thank you once again for your time and constructive feedback.
Reviewer 3 Report
Comments and Suggestions for Authors
The authors have addressed all my comments, resulting in a significantly improved manuscript. The conclusion effectively summarizes the findings but could be more concise. The final sentence should be more action-oriented to emphasize the study’s implications. A proofreading is recommended to ensure consistency. Keywords may be arranged in alphabetical order. Some sentences could be simplified for clarity, particularly in the discussion and conclusion. For example, "The data were collected through a questionnaire hosted on an online platform" could be revised to "Data were collected through an online questionnaire."
Author Response
Dear Reviewer 3,
We sincerely appreciate your review, constructive feedback and your positive feedback on the English language, as it was revised by native professional speakers. We are pleased to hear that you find the manuscript significantly improved and that the introduction, methods, results, and conclusions are now appropriate.
Following your suggestions, we have refined the conclusion to make it more concise and action-oriented, better emphasizing the study’s implications. We have also conducted a final proofreading to ensure consistency throughout the text. Additionally, the keywords have been reordered alphabetically.
Furthermore, as you recommended, we have simplified certain sentences for greater clarity, particularly in the discussion and conclusion sections. As requested, we have revised phrasing such as "The data were collected through a questionnaire hosted on an online platform" to "Data were collected through an online questionnaire."
Thank you once again for your insightful comments, which have greatly contributed to improving the manuscript.
Reviewer 4 Report
Comments and Suggestions for Authors
Dear Authors,
Thank you for your detailed responses to my comments. I appreciate the effort you have made in addressing the suggestions for improving clarity, coherence, and analytical depth.
Regarding the reliability of the SHE rapid assessment tool, your response indicates that no publicly available information exists on its reliability. Given the importance of ensuring methodological rigor, I suggest the following:
- Clarify whether any prior studies have assessed the reliability of this tool – If previous research using the SHE rapid assessment tool has reported reliability measures (e.g., Cronbach’s alpha), citing these studies would help support the validity of your instrument.
- Acknowledge this as a limitation – If no reliability data are available, this should be explicitly stated as a limitation in the discussion or methodology, as it affects the strength of the conclusions drawn from the results.
- Provide internal consistency data – Since your study includes a large sample (927 participants), it would be valuable to calculate and report Cronbach’s alpha for the items used. This would offer empirical support for the reliability of the tool within the context of international schools and strengthen the credibility of your findings.
Without an assessment of the tool’s reliability, the study lacks evidence of its measurement consistency, which is a crucial aspect of research validity. If reliability data cannot be provided, it is essential to explicitly state that the study does not establish the tool’s reliability and discuss the implications this may have for interpreting the results.
I look forward to your clarification on this matter. Thank you again for your thorough revisions and for engaging in this constructive dialogue.
Best regards
Author Response
Dear reviewer 4,
We sincerely appreciate your thoughtful review and constructive feedback. We are pleased to hear that you find the manuscript significantly improved, and that the revisions made to the introduction and methods have enhanced their clarity, coherence, and analytical depth.
Regarding your suggestion about the reliability of the SHE rapid assessment tool, we acknowledge that no publicly available information exists on its reliability. As per your recommendation, we have explicitly stated this as a limitation in the discussion section. Although the tool is widely used and recommended by the SHE network, a respected and noteworthy institution supported by WHO EURO, we recognize that the absence of published reliability data may affect the strength of the conclusions drawn from the results.
Since the validation of the tool was not an objective of our study, we did not develop methods or analyses to calculate and report Cronbach’s alpha for the items used at this stage of the study. However, we agree that calculating internal consistency would offer empirical support for the reliability of the tool within the context of international schools and could strengthen the credibility of our findings. We have noted this in the limitation section and greatly appreciate your suggestion, which we will consider for future research.
Thank you again for your valuable insights and constructive dialogue.